# Iflavirus Covert Infection Increases Susceptibility to Nucleopolyhedrovirus Disease in *Spodoptera exigua*

**DOI:** 10.3390/v12050509

**Published:** 2020-05-05

**Authors:** Arkaitz Carballo, Trevor Williams, Rosa Murillo, Primitivo Caballero

**Affiliations:** 1Institute for Multidisciplinary Research in Applied Biology, Universidad Pública de Navarra, 31006 Pamplona, Spain; arkaitz.carballo@unavarra.es (A.C.); pcm92@unavarra.es (P.C.); 2Departamento de Biotecnología, Agronomía y Alimentos, Universidad Pública de Navarra, 31006 Pamplona, Spain; 3Instituto de Ecología AC, Xalapa, Veracruz 91073, Mexico; trevor.inecol@gmail.com

**Keywords:** iflavirus, SeMNPV, covert infection, *Spodoptera exigua*

## Abstract

Naturally occurring covert infections in lepidopteran populations can involve multiple viruses with potentially different transmission strategies. In this study, we characterized covert infection by two RNA viruses, Spodoptera exigua iflavirus 1 (SeIV-1) and Spodoptera exigua iflavirus 2 (SeIV-2) (family Iflaviridae) that naturally infect populations of *Spodoptera exigua*, and examined their influence on susceptibility to patent disease by the nucleopolyhedrovirus Spodoptera exigua multiple nucleopolyhedrovirus (SeMNPV) (family Baculoviridae). The abundance of SeIV-1 genomes increased up to ten-thousand-fold across insect developmental stages after surface contamination of host eggs with a mixture of SeIV-1 and SeIV-2 particles, whereas the abundance of SeIV-2 remained constant across all developmental stages. Low levels of SeIV-2 infection were detected in all groups of insects, including those that hatched from surface-decontaminated egg masses. SeIV-1 infection resulted in reduced larval weight gain, and an unbalanced sex ratio, whereas larval developmental time, pupal weight, and adult emergence and fecundity were not significantly affected in infected adults. The inoculation of *S. exigua* egg masses with iflavirus, followed by a subsequent infection with SeMNPV, resulted in an additive effect on larval mortality. The 50% lethal concentration (LC_50_) of SeMNPV was reduced nearly 4-fold and the mean time to death was faster by 12 h in iflavirus-treated insects. These results suggest that inapparent iflavirus infections may be able to modulate the host response to a new pathogen, a finding that has particular relevance to the use of SeMNPV as the basis for biological pest control products.

## 1. Introduction

Covert virus infections of insects persist without obvious signs of disease and are usually non-lethal [1,2,3,4,5,6,7,8]. As covertly infected hosts often survive and reproduce, the covert infection strategy is frequently associated with opportunities for vertical transmission [9,10]. Covert infection involves a low level of virus replication and a low level of transcriptional activity from the viral genome [11,12]. The pathogen avoids clearance by the host immune response by a variety of mechanisms, including microRNA interference [13,14].

Complex interactions can arise when more than one virus infects a shared host [15,16,17]. On a cellular level, the capacity to sustain persistent viral co-infections may be a common characteristic of animal cells, including those of invertebrates [18]. Virus–virus interactions in co-infected hosts may involve: (a) direct interactions among genes or gene products of the viruses; (b) indirect interactions through alteration of the host environment, and (c) indirect interaction by modulation of both innate and induced host immune responses [19].

The Spodoptera exigua multiple nucleopolyhedrovirus (SeMNPV), is the active ingredient of several virus-based insecticides used for the biological control of this pest [20]. This virus is highly pathogenic and is usually transmitted when viral occlusion bodies (OBs) are release from virus-killed larvae and are subsequently consumed by conspecific larvae feeding on OB-contaminated plant material [21]. Alternatively, SeMNPV can produce covert infections with measurable effects on host growth and reproduction [4]. Such infections can be transmitted from parents to their offspring over several generations [17].

Iflaviruses are positive ssRNA viruses that cause lethal infections in silkworms, honey bees and a number of other invertebrates [22]. Iflaviruses often produce non-lethal persistent infections in their hosts, including species of Lepidoptera [23,24,25,26]. During transcriptome studies, two iflaviruses were discovered in individuals of the armyworm *Spodoptera exigua* that showed no apparent signs of disease. These were named Spodoptera exigua iflavirus 1 (SeIV-1) [27] and Spodoptera exigua iflavirus 2 (SeIV-2) [28]. Both these viruses were subsequently detected individually and in mixed infections with SeMNPV in populations of *S. exigua* in southern Spain [17], indicating that these viruses are likely to interact frequently in natural populations of this pest. 

Simultaneous co-inoculation of larvae with mixtures of SeMNPV OBs, and SeIV-1, and SeIV-2 particles significantly affected the insecticidal properties of SeMNPV [29]. Evidence was also obtained for the physical association of iflavirus particles within OBs produced in co-infected insects, underscoring the intimate nature of the interactions that are likely to occur among these different types of viruses in nature [30]. However, to date, no studies have been conducted on the nature of covert infection by iflaviruses in *S. exigua*, and the consequences of pre-existing covert iflavirus infection on insect susceptibility to SeMNPV. Consequently, in the present study, we initially determined the abundance of SeIV-1 and SeIV-2 in a reference infected *S. exigua* population. We then performed inoculation trials, monitored virus loads within covertly infected hosts and measured the sublethal effects of covert infection on host growth and reproduction. Finally, we examined the susceptibility of iflavirus-infected insects to lethal doses of SeMNPV OBs and the progression of the lethal disease.

## 2. Materials and Methods 

### 2.1. Insect and Virus Stock

The *Spodoptera exigua* colony used in the experiments was originally obtained from Andermatt Biocontrol (Grossdietwil, Switzerland). Insects were maintained in the insectary facilities of the Universidad Pública de Navarra. Larvae were reared on diet [31] at 25 ± 2 °C, 50% ± 10% relative humidity and a 16:8 h light:dark photoperiod. When a persistent iflavirus infection occurred in this insect colony, the insects were reared separately in disinfected bioclimatic chambers used exclusively for this purpose.

Spodoptera exigua multiple nucleopolyhedrovirus was the Spanish isolate SeMNPV-SP2 [32]. Viral OBs were amplified in iflavirus-free *S. exigua* larvae and consistently proved negative for the presence of SeIV-1 and SeIV-2 by reverse transcription quantitative PCR (RT-qPCR), as described previously [33].

SeIV-1 and SeIV-2 particles were isolated from *S. exigua* larvae from a colony that had been confirmed to be persistently infected [29]. For this, the gut tissues from 200 fourth instars were dissected, lyophilized and homogenized in 0.01 M potassium phosphate buffer (pH 7.4) with 0.45% (*w/v*) diethyldithiocarbamic acid (DIECA) and 0.2% (*v/v*) β-mercaptoethanol (2.5 mL of buffer per gram of larval tissue), sonicated for 20 s and filtered through two layers of cheesecloth. The filtrate was loaded on to 10%, 30% and 60% discontinuous sucrose gradient and centrifuged for 6 h at 60,000× *g* at 10 °C. Virus particles were collected from fractions between 30% and 60% sucrose and RNA was extracted using RNAzol^®^ RT (Sigma-Aldrich, St. Louis, MO, USA) following the manufacturer’s protocol. Quantitative PCR (qPCR) was undertaken as described in Section 2.3 to determine the concentration of SeIV-1 and SeIV-2 genomes in each sample. For this study, a mixed inoculum of SeIV-1 + SeIV-2 was prepared, in which both iflaviruses were present in equal proportions (50% SeIV-1 + 50% SeIV-2).

### 2.2. Establishment and Time-Course of Iflavirus Infection

The establishment and time-course of iflavirus infection were studied to determine whether the abundance of iflavirus changed during insect development. Host egg masses were immersed in iflavirus suspension to emulate the acquisition of infection by neonate larvae during egg hatching. Inoculated insects were then reared through to the adult stage and samples were taken and analyzed for the presence of SeIV-1 and SeIV-2 by qPCR (see Section 2.3).

For this, egg masses were initially disinfected in 0.5% sodium hypochlorite for 10 min, thoroughly rinsed in water, and allowed to dry. Disinfected egg masses were then placed in one of three concentrations of SeIV-1 + SeIV-2 suspension containing 10^7^, 10^8^, and 10^9^ total viral genomes/µL for 5 min. Control egg masses were either untreated (not treated with hypochlorite solution or virus suspension, named non-treated NT) or were treated with hypochlorite solution alone followed by immersion in water instead of virus (named the NaClO treatment). Egg masses (~50 eggs each) of *S. exigua* were inoculated with each treatment and each placed in a 100-cm^3^ plastic container with a piece of diet until hatching. Four-day-old larvae were individualized in 24-compartment plates (Corning, New York, NY, USA) provided with diet and reared to the adult stage in the rearing conditions described in Section 2.1. To test for the presence of viruses, three individuals were randomly selected at the second, third, fourth and fifth instars, pupal and adult stages. These samples were frozen at −20 °C and were then used for total RNA extraction and iflavirus quantification as describe below. The entire experiment was performed three times.

### 2.3. Virus Detection and Quantification

For the detection of covert iflavirus infection, total RNA was isolated from insect tissues using the Master Pure Complete RNA Purification kit (Epicentre Biotechnologies Corp., Madison, WI, USA). One of the three following insect samples: (i) entire larva, (ii) pupa or; (iii) dissected adult abdomen, was placed individually in a 2-mL microfuge tube with sterile ceramic beads, 300 µL tissue lysis solution and 1 µL proteinase K (50 ng/µL). Samples were homogenized using MP FastPrep-24 tissue cell homogenizer (MP Biomedicals, Irvine, CA, USA) at 4 m/s for 20 s and incubated at 65 °C for 15 min in constant agitation. For RNA extraction, protein precipitation reagent was added, centrifuged at 10,000 × *g* for 10 min and the RNA was precipitated with isopropanol. The resulting nucleic acid pellets were treated with RNAse-free DNAse buffer and 5 µL of DNAse for 30 min at 37 °C. A volume of 200 µL of 2 × T and C lysis solution was added, vortexed for 5 s and 200 µL of protein precipitation reagent was then added and vortexed for 10 s. Debris was pelleted by centrifugation and the supernatant was precipitated once with isopropanol and twice with 70% ethanol. Finally, RNA was resuspended in 20 µL of DEPC-treated water and stored at −20 °C. Blank extraction samples containing only water were processed in parallel to control for contamination during the extraction process. All equipment and reagents were previously sterilized and treated with DEPC to remove RNases.

To quantify viral loads in *S. exigua*, reverse transcription quantitative PCR (RT-qPCR) was performed for SeIV-1 and SeIV-2 separately using virus-specific primers that had been designed previously for this purpose [29,33] (Appendix A). The PCR amplification cycle was as follows: 3 min at 95 °C, 10 s at 95 °C and 30 s at 62 °C for 45 cycles followed for a melting curve of 5 s per 0.5 °C from 65 to 95 °C. Total cDNA was used as template to detect each of the iflaviruses. To obtain cDNA, 1 µg of RNA was reverse transcribed using SuperScript II Reverse Transcriptase (Promega, Madison, WI, USA). The reverse transcription mix consisted of 2 µL of 5× buffer (Promega, Madison, WI, USA), 1.2 µL MgCl_2_ (25 mM), 0.5 µl dNTP mix (10 mM), 0.8 µL DEPC-treated water and 1 µL ImProm-II reverse transcriptase (Promega, Madison, WI, USA). The mixture was added to RNA samples and incubated at 25 °C for 5 min, followed by 42 °C for 60 min and 70 °C for 15 min. A qPCR reaction with SYBR Green was then performed in a CFX96 Touch™ Real-Time PCR Detection System (Bio-Rad, Hercules, CA, USA) in 96-well plates. A 9 µL volume of mastermix containing 5 µL SYBR Green, 0.5 µL of each of the primers (10 µM) and 3 µL water was added to 1 µL of cDNA template. For the construction of standard curves, the PCR products generated with specific primers for SeIV-1 and SeIV-2 were cloned into a pGEM^®^-T Easy cloning vector (Promega, Madison, WI, USA). Plasmid DNA were quantified in a UV spectrophotometer (Eppendorf BioPhotometer Plus, Hamburg, Germany), and eight-fold serial dilutions in sterile MilliQ water (1 × 10^−1^ to 1 × 10^−7^ ng/µL) were used in duplicate as standards for the construction of a standard curve. Data were acquired and analyzed using Bio-Rad CFX Manager 3.1 software (Bio-Rad, Hercules, CA, USA). The regression coefficient of determination of the standard curves exceeded R^2^ = 0.95 in both cases, with an efficiency between 90% and 110% [34]. In all cases, the lowest reference concentration, 10^−7^ ng/µL, represented the limit of detection and showed correct amplification curves and the expected melting temperatures of 77.0 and 79.5 °C for SeIV-1 and SeIV-2, respectively. The corresponding Ct values were assigned as cut-off points for each virus, so that higher values were considered to be virus-free samples. Ct values among different insect-stage samples were normalized using the corresponding values of the reference gene *ATP-synthase subunit C,* as previously reported [33] (Appendix A). The corresponding viral titers (genomes/µL) for SeIV-1 and SeIV-2 were normalized by log transformation and subjected to two-way analysis of variance (ANOVA) with virus treatment and host stage as factors. Mean separation was performed by Tukey test in SPSS (v. 25, 2017 IBM, Armonk, NY, USA). The relationship between SeIV-1 and SeIV-2 loads per individual was examined by Spearman’s rank correlation (SPSS, v. 25, 2017 IBM, Armonk, NY, USA).

### 2.4. Sublethal Effects of SeIV-1 Infection

The effects of covert infection by SeIV-1 on the development and reproduction of *S. exigua* were determined. For this, egg masses were disinfected using hypochlorite solution washed and immersed in a suspension of 1 × 10^8^ SeIV-1 genomes/µL, as described in the time-course experiment. A group of 24 larvae that emerged from SeIV-1-inoculated eggs were individually reared to adulthood. A control group of 24 mock-infected larvae were treated identically. Three independent replicates were performed using different batches of eggs. Larval weight and instar were recorded at 6 days after egg hatching. Pupae were sexed, weighted and examined daily for adult emergence. Adults from each group were randomly selected to evaluate adult fertility. For this, five females and five males from each treatment and replicate were placed in groups and allowed to mate in paper bags with continuous access to water in small containers with a cotton wick. The number of eggs laid on the interior of the paper bag was counted for the following 6 days. Larval weights, pupal weights, and development time (egg hatching to pupation), were not normally distributed and were compared by Mann–Whitney U test. The percentage of adults that emerged, the sex ratio and the fecundity of each group of female moths were each subjected to one-way ANOVA (SPSS, v. 25, 2017 IBM, Armonk, NY, USA).

### 2.5. Effects of Iflavirus Infection on SeMNPV Insecticidal Properties

In order to determine whether pre-existing covert infection by iflavirus affected the susceptibility of *S. exigua* larvae to lethal infection by SeMNPV OBs, or the progression of lethal disease, a bioassay was performed in which two egg masses of *S. exigua* were treated with 10^8^ viral genomes/µL of SeIV-1 + SeIV-2, or water (mock-inoculated control), as described above. Groups of 30 *S. exigua* first instars that had hatched from treated egg masses were starved overnight, and once molted to the second instar, larvae were allowed to drink from one of five OB suspensions containing 2.54 × 10^5^, 8.18 × 10^4^, 2.72 × 10^4^, 9.09 × 10^3^ and 3.03 × 10^3^ OBs/mL, by the droplet-feeding method [11]. This range of concentrations had previously been demonstrated to kill between 95% and 5% of inoculated larvae. Groups of 30 larvae, both SeIV-1+SeIV-2-inoculated and control insects, were treated identically, except that the feeding solution did not contain OBs. Larvae that drank the OB suspension within 10 min were placed individually in 24-well plates (Corning, New York, NY, USA) containing artificial diet. Inoculated larvae were maintained at 25 ± 1 °C, 50% ± 5% RH and SeMNPV-induced mortality was recorded daily for 7 days post-inoculation, except in the case of the larvae that consumed the highest concentration of OBs, for which mortality was recorded at 8-h intervals. Larvae were consider to have died when they did not respond to mechanical stimuli. The experiment was performed three times.

OB production was determined for larvae treated with the highest inoculum concentration by placing dead larvae individually in 1.5 mL vials. Ten virus-killed larvae were randomly selected per replicate for estimation of numbers of OB/larva. Each cadaver was homogenized individually in 1 mL of distilled water and OBs were counted in triplicate using a Neubauer hematocytomer (Hawksley, Lancing, UK) and a phase contrast microscope at 400×.

Concentration–mortality results were subjected to Probit regression analysis using the POLO-PC program [34]. Time–mortality data recorded for larvae treated with the highest inoculum concentration were subjected to Weibull survival analysis in GLIM4 [35]. The validity of the Weibull model was determined by comparing fitted values with the Kaplan–Meier survival function estimated values. OB production per insect values were averaged for each replicate and then subjected to ANOVA (SPSS, v. 25, 2017 IBM, Armonk, NY, USA).

## 3. Results

### 3.1. Time-Course of Iflavirus Infections

Control insects (NT) and those treated with hypochlorite solution in the egg stage were found to harbor low levels of infection by both iflaviruses (Figure 1a,b; Appendix A). Insects obtained from colonies of *S. exigua* held in other European research institutions were also found to harbor iflaviruses, despite rigorous measures taken to avoid cross-contamination among insect colonies in our laboratories, including the use of different people and different rooms used for the rearing of each colony. SeIV-1 treatment at concentrations of 10^7^–10^9^ genomes/µL resulted in a 100- to 10,000-fold increase in viral titers when compared to control groups (NT and hypochlorite-treated) across all immature stages (*F*_4,74_ = 22.039; *p* <0.001), whereas, in the adult stage, SeIV-1 titers were similar (~10^8^ genomes/µL) in insects from all treatment groups (Figure 1a). SeIV-1 titers did not differ significantly among the different stages over the course of insect development (*F*_5,74_ = 2.026; *p* = 0.094). The interaction of the insect stage and virus treatment was not significant (*F*_20,74_ = 0.594; *p* = 0.896).

In contrast, SeIV-2 titers were very similar across the different treatments (Figure 1b), including the no-treatment (NT) control and the hypochlorite treatment (*F*_4,84_ = 0.330; *p* = 0.857). SeIV-2 titers did not differ among developmental stages except in the adult stage in which a significant decrease was observed (*F*_5,84_ = 6.854; *p* < 0.001). The interaction of these factors was not significant (*F*_20,84_ = 0.259; *p* = 0.999). SeIV-2 titers varied from 10^4^ to 10^5^ genomes/µL in most cases and were consistently lower than those of SeIV-1, except in the case of the no-treatment controls and hypochlorite treatment that were similar in SeIV-1- and SeIV-2-infected insects (Figure 1a,b) No evidence of any correlation in the quantities of each virus in individual insects was detected (data pooled for treatment and stages), as viral loads varied independently (Spearman’s *r* = 0.161; *n* = 90; *p* = 0.129).

To control for host growth effects, SeIV-1 and SeIV-2 titers were normalized with reference to *ATP-synthase* gene expression (Figure 1c). SeIV-1 expression exceeded the reference gene expression levels in all iflavirus treatments except for the NT control or hypochlorite treatment insects, indicating a positive relationship between viral replication and developmental stage. SeIV-2 replication did not exceed that of the reference gene in any development stage, and was independent of inoculum concentration, so that the relative abundance of SeIV-2 was almost constant across treatments and developmental stages. The following studies therefore focused on sublethal effects of SeIV-1 in the presence of a background persistent infection by SeIV-2.

### 3.2. Sublethal Effects of SeIV-1 Infection

The median larval weight at 6 days post-inoculation was approximately 30% lower in the SeIV-1 treated group compared to mock-infected insects (Mann-Whitney *U* = 1548.0; *p* = 0.048) (Table 1). Surprisingly, this marked difference in growth was not carried over to the pupal stage in which median weights were similar in both groups (Mann–Whitney *U* = 818.0; *p* = 0.593). Development time, measured from egg hatch to pupation did not differ significantly between treatments at approximately 17 days in both cases (Mann–Whitney *U* = 3.0; *p* = 0.507). The percentage of adults that emerged was also similar between control and treated insects (*F*_1,4_ = 0.333; *p =* 0.595), whereas sex ratio in the iflavirus treatment was biased in favor of males by 15.7% compared to control insects (*F*_1,4_ = 8.472; *p =* 0.044). Mean fecundity did not differ significantly between SeIV-1-treated and control females that were held in groups of five pairs (*F*_1,4_ = 2.958; *p =* 0.161).

### 3.3. Effects of Iflavirus Infection on SeMNPV Insecticidal Properties

The susceptibility of second instars to SeMNPV OBs significantly increased in larvae that hatched from SeIV-1 + SeIV-2-treated egg masses (Table 2). The LC_50_ value of SeMNPV OBs was reduced by ~4-fold for the iflavirus treated insects when compared to the control insects. No mortality was registered in the mock-infected larvae of either insect groups.

The speed of kill of SeMNPV was significantly affected by iflavirus treatment, resulting in a 12-h reduction in the mean time to death in insects that hatched from iflavirus-treated egg masses (Table 2) compared to the control group (Weibull analysis, *t* = 5.16; df = 94; *p* < 0.05). Mean (±SE) OB production per larva did not differ significantly in control larvae (2.7 × 10^7^ ± 6.4 × 10^6^ OBs/larva) compared to larvae from the iflavirus treatment (3.9 × 10^7^ ± 9.9 × 10^5^ OBs/larva) (*F*_1,4_ = 1.26; *p* = 0.34).

## 4. Discussion

Covert infections by iflaviruses appear to be common in *S. exigua* colonies from different origins [33], including field-caught and captive-reared individuals [17]. In this study, the inoculation of *S. exigua* egg masses with different concentrations of SeIV-1 and SeIV-2 resulted in virus-specific infection loads that varied during host development. For all concentrations tested, SeIV-1 loads dramatically increased up to 10,000-fold in iflavirus-treated groups, proving that the pathogen replicated continuously during insect development. In contrast, SeIV-2 loads did not increase from low persistent virus levels when eggs were inoculated, even at an inoculum concentration of 10^9^ genomes/µL. Indeed, adult loads of SeIV-2 decreased compared to those present in the larval stage, suggesting a limited amplification of this virus. Taken together, these results indicate that SeIV-1 successfully established a productive infection, despite the presence of a simultaneous persistent infection by SeIV-2.

The sublethal effects of covert iflavirus infection were measured only for the SeIV-1, since we confirmed that viral inoculation resulted in increased loads only in this virus. SeIV-1-treated insects exhibited reduced larval weight gain and a reduction in the proportion of adult female insects. Reduced larval weight gain was not reflected in pupal weight and appeared not to influence female fecundity, in contrast to the sublethal effects typically observed in baculovirus-infected insects [4]. Additionally, our findings suggest that iflavirus inoculation might adversely affect population growth by reducing the production of females. Previously, SeIV infections had not been reported to be detrimental to the host, and could pass unnoticed because they are usually not lethal [27,33]. Another iflavirus, the deformed wing virus (DWV), does not cause honeybee death but has deleterious effects on honeybee foraging and survival, which can contribute to colony declines [2]. Similarly, Helicoverpa armigera iflavirus (HaIV) delayed development and reduced survival in larval stage insects [36].

The magnitude of the pathological effects induced by iflavirus infections has been related to the viral load in the host [37]. Low-level iflavirus infections are often associated with no obvious signs of disease, whereas high levels may result in measurable pathological effects and host death [38,39]. For example, when low-level persistent DWV infections in bees were vectored by Varroa mites, virus titers were amplified 1000-fold and virulent disease occurred in affected colonies [38]. Similarly, asymptomatic bees were shown to harbor sacbrood virus, though at significantly lower levels than those observed in individuals showing signs of sacbrood disease [39].

Iflavirus infections altered host susceptibility to lethal infection by SeMNPV and may be expected to alter the efficacy of biological insecticides based on this pathogen. Covert infection by SeIV-1 induced by prior egg mass inoculation increased SeMNPV-induced mortality and the speed of kill. Similar effects were observed when SeMNPV OBs and SeIV-1 + SeIV-2 particles were simultaneously inoculated in *S. exigua* larvae [29]. The changes in host susceptibility may be a result of iflavirus-induced modification in host defenses, such as the activation of innate antiviral immunity responses [40], apoptosis [41] and RNA interference activation as a cellular defense mechanism [42]. Iflavirus infections have been observed to activate host immune responses by up-regulating antimicrobial peptides and down-regulating the expression of phenoloxidase genes that form part of the insect humoral immune response [38]. The determination of the mechanisms through which iflaviruses interact with one another, and through which they affect host susceptibility to superinfection by other viruses, will require detailed future studies.

In the present study, low loads of iflavirus were detected in the untreated group (NT), indicating a persistent low-level infection of the *S. exigua* colony. An increasing number of studies evidence the prevalence of iflavirus infections in lepidopterans and other insect populations, which are often detected serendipitously during new generation sequencing or transcriptomic studies [43]. Lepidopteran iflaviruses have been reported from a diversity of species across a wide geographical range, such as the processionary moth *Thaumetopea pityocampa* [44], the gypsy moth *Lymantria dispar* [23], the oak silkmoth *Antheraea pernyi* [24], the palm tree moth *Opsiphanes invirae* [25], and the neotropical butterfly *Heliconius erato* [45]. In none of these cases did the infected insects show overt signs of disease.

Iflavirus virulence (in terms of proliferation in host cells) has been related to the route of transmission of these viruses, for example, depending on the route of transmission, DWV can cause either symptomatic infection, characterized by high levels of virus, or covert infection, involving low levels of virus replication [37]. Vertically transmitted iflavirus infections do not involve high levels of replication and do not produce overt signs of disease [46]. This dual transmission system is frequently reported among iflavirus groups and it may be applicable to SeIV-1 and SeIV-2. In a previous study, we observed a rapid spread of SeIV-1 infection in insect colonies [33], likely mediated by food contamination, as reported in HaIV [26]. Another study revealed SeIV-1 and SeIV-2 transgenerational transmission detected in 20%–40% of the offspring from field-caught adults [17]. As we did not observe any reduction in virus titers following surface decontamination of egg masses, the persistent infection detected in early instars might be due to transovarial (within-egg) rather than transovum (on the outer surface of the egg) transmission. Similarly, HaIV infections were less prevalent by vertical (28%) compared to horizontal transmission (75%) [26].

Here, we observed divergent trends between the iflavirus species during host development. SeIV-1 showed high levels of transcription, suggesting high levels of replication and opportunities for horizontal transmission, through the production of iflavirus contaminated feces or regurgitation that contaminate food eaten by conspecific insects. In contrast, the reduced loads and limited amplification of SeIV-2 suggest persistent and relatively benign infection and potential opportunities for vertical transmission. In this sense, the iflavirus of *Lymantria dispar* (LdIV-1) was capable of producing persistence infections in cell culture as a result of the viral suppression of cell-mediated RNA silencing [23].

Considering the high prevalence of iflavirus infections in Lepidoptera, it seems likely that these viruses could potentially improve the efficacy of SeMNPV-based insecticides targeted at the control of *S. exigua* infestations in field and greenhouse crops. In addition, the deleterious sublethal effects of SeIV-1 infections on larval growth and adult sex ratio might contribute to pest suppression. However, the biosecurity implications of the presence of iflaviruses in biological insecticides and the insect colonies used to produce these products remain unclear, and this is an issue that merits further study.

## Figures and Tables

**Figure 1 viruses-12-00509-f001:**
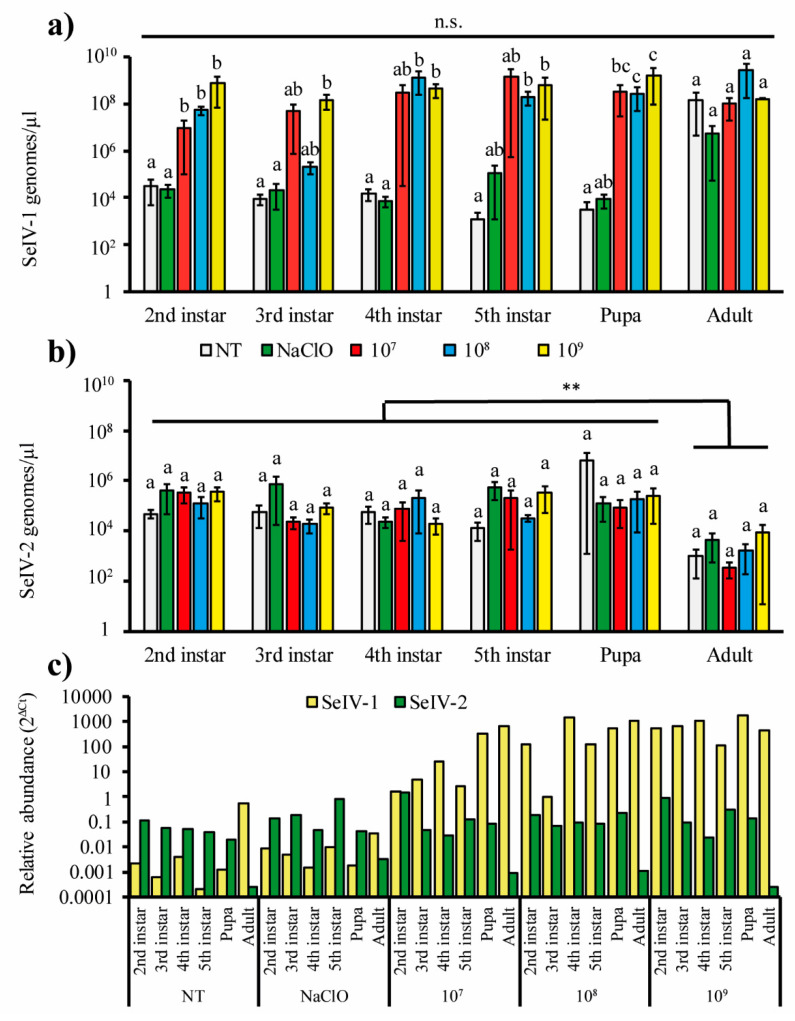
Quantification of titers of (**a**) Spodoptera exigua iflavirus 1 (SeIV-1) and (**b**) SeIV-2 in the developmental stages of *Spodoptera exigua*. Viral loads were estimated by quantitative PCR (qPCR) for insects that hatched from egg masses that had been treated with 10^7^, 10^8^ and 10^9^ genomes/µL of a mixture of SeIV-1 and SeIV-2 (time-course experiment). The control egg masses were not treated (NT) or treated with hypochlorite solution (NaClO). Vertical bars indicate SE. Columns headed by different letters differ significantly for comparisons of treatments within each developmental stage (ANOVA, Tukey *p* < 0.05). Horizontal lines indicate comparisons among developmental stages in (**a**) and (**b**). n.s. indicates not significant, whereas ** indicates significant difference (ANOVA, *p* < 0.01). (**c**) Relative abundance (mean 2^ΔCt^ values) of SeIV-1 and SeIV-2 in insect samples calculated with reference to Ct values for the *ATP-synthase* reference gene.

**Table 1 viruses-12-00509-t001:** Sub-lethal effects of SeIV-1 infection on *S. exigua* in terms of the larval weight (at 144 hpi), pupal weight, development time to pupa, adult emergence, sex ratio and fecundity of groups of five females held together.

Treatment Group	Median Larval Weight (mg) ^1^	Median Pupal Weight (mg) ^1^	Median Development Time to Pupae (Days) ^1^	Mean Adult Emergence ^2^ ± SE (%)	Mean Sex Ratio ±SE (% Female) ^2^	Mean Fecundity ±SE (Eggs/Group of Females) ^2^
Control	170.0a (134; *n* = 68)	98.0a (35; *n* = 45)	17.3a (12; *n* = 31)	76.6 ± 5.3a (*n* = 43)	54.8 ± 2.9a (*n* = 41)	1203 ± 427.5a (*n* = 15)
SeIV-1	117.0b (130; *n* = 58)	99.0a (21; *n* = 39)	17.4a (12; *n* = 26)	72.0 ± 5.9a (*n* = 38)	39.1 ± 4.6b (*n* = 36)	446 ± 104.5a (*n* = 15)

^1^ Different letters indicate significant differences between virus-treatment groups by Mann–Whitney test (*p* < 0.05). Interquartile range and numbers of insects measured are indicated in parenthesis. ^2^ Different letters indicate significant differences between virus-treatment groups by one-way ANOVA (*p* < 0.05). Numbers of insects are indicated in parenthesis.

**Table 2 viruses-12-00509-t002:** LC_50_ values, relative potency and mean time to death (MTD) for *S. exigua* second instars inoculated with SeMNPV viral occlusion bodies (OBs) on either a mock-infected control group or a group that hatched from SeIV-1 + SeIV-2-treated egg masses.

Treatment	LC_50_ ^1^ (×10^4^ OBs/mL)	Relative Potency	95% Fiducial Limits		χ^2^	MTD ^2^ (h)	95% Fiducial Limits	
			Low	High			Low	High
Control	5.87	1.00	-	-	5.27	90.77a	88.83	92.74
SeIV-1 + SeIV-2	1.56	3.76	2.23	6.35	5.93	78.93b	77.62	80.26

^1^ A test for non-parallelism was not significant (*χ*^2^ = 2.56; df = 1; *p* = 0.11), so that regressions were fitted with a common slope of 0.989 ± 0.116 (mean ± S.E.). ^2^ Mean time to death values calculated for insects that consumed the highest concentration of SeMNPV OBs. MTD values labeled with different letters differed significantly (*t*-test, *p* < 0.05).

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
