# Peer review of "Iflavirus Covert Infection Increases Susceptibility to Nucleopolyhedrovirus Disease in Spodoptera exigua"

_viruses, 2020, doi:10.3390/v12050509_

Round 1

Reviewer 1 Report

The paper presents the results of a very thorough and carefully controlled set of experiments to characterise covert infection by two iflaviruses - SeIV-1 and SeIV-2 that are found in naturally occurring populations of Spodoptera exigua.  The results, which are very well presented and described, demonstrated that inoculation of eggs with a mix of SeIV-1  and SeIV-2 results in a significant increase in SeIV-1 but not SeIV-2 across all developmental stages.  SeIV-2 was found at low-levels in control insects.  When inoculated eggs were also infected with a bacuovirus (SeMNPV) that was an additive effect on mortality - faster time to death and reduced LC50 compared to infection without iflavirus.

The experiments are clearly described and very well presented so I do not have any questions on the data except one minor point - in the methods it says eggs are inoculated with 108 genomes/ul but in the results e.g. Figure 1 A reference is made to inoculation with 107, 108 or 109 genomes/ul.  Can this be clarified.

Could the authors add to the discussion their thoughts on whether they think the fact that the S. exigua insects used had a covert infection of SeIV-2, had any impact on the ability of the SeIV-1 to replicate but not the SeIV-2?  Or is there some other reason to explain the different results for each virus - I couldn't see this specifically in the discussion.  Have the authors tried inoculating eggs with SeIV-1 or -2 singly?  What happens - does SeIV-1 infection still increase but SeIV-2 not?  I am not asking for this to be done, just whether the authors have previously done this work.

One typo spotted - line 67 nature of covert .....

Author Response

The experiments are clearly described and very well presented so I do not have any questions on the data except one minor point - in the methods it says eggs are inoculated with 108 genomes/ul but in the results e.g. Figure 1 A reference is made to inoculation with 107, 108 or 109 genomes/ul.  Can this be clarified.

Response 1: We performed three different experiments. For the time-course experiment (experiment 1) three different concentrations of iflaviruses were used (line 107). In contrast the studies on sublethal effects (experiment 2) and susceptibility to SeMNPV (experiment 3), a single concentration (10e8 genomes/µl) was used. This is stated in the Methods. For clarity, a phrase has been added in to the legend of Figure 1 (line 239-240).

Could the authors add to the discussion their thoughts on whether they think the fact that the S. exigua insects used had a covert infection of SeIV-2, had any impact on the ability of the SeIV-1 to replicate but not the SeIV-2?  Or is there some other reason to explain the different results for each virus - I couldn't see this specifically in the discussion.  Have the authors tried inoculating eggs with SeIV-1 or -2 singly?  What happens - does SeIV-1 infection still increase but SeIV-2 not?  I am not asking for this to be done, just whether the authors have previously done this work.

Response 2: In this study, we did not focus on interactions between each of the iflaviruses, and we did not examine single virus infection in any experiment. However, subsequent experiments by our colleague (Salvador Herrero, Universidad de Valencia, Spain), as part of a collaborative research project, have addressed these issues and these will be submitted to a virological journal in the near future. The aim of the present study was to determine the impact of iflavirus inoculation and the influence of pre-existing covert infection on insect susceptibility to baculovirus infection.  No changes made.

One typo spotted - line 67 nature of covert .....

Response 3: Done.

Reviewer 2 Report

In this manuscript “Iflavirus covert infection increases susceptibility to nucleopolyhedrovirus disease in Spodoptera exigua”, Carballo et al. have characterized covert infection by two RNA viruses, Spodoptera exigua iflavirus 1 (SeIV-1) and Spodoptera exigua iflavirus 2 (SeIV-2), which naturally infect Spodoptera exigua populations, and examined their influence on susceptibility to SeMNPV. The authors provide a set of well-organized experimental data showing that after the host egg surface has been contaminated with a mixture of SeIV-1 and SeIV-2 particles, the abundance of SeIV-1 increased sharply, whereas that of SeIV-2 remained constant at all developmental stages. Inoculation of S. exigua egg masses with iflavirus, followed by SeMNPV infection, has a cumulative effect on larval mortality, suggesting that unapparent iflavirus infection may modulate the host response to a new pathogen.

Comments:

  1. Although these experiments are well-designed and the authors provide reasonable conclusions, the results do not significantly increase our knowledge of virus/host interactions or virus/virus competition in hosts.
  2. What are the competitive mechanisms between SeIV-1 and SeIV-2, and what is the mechanism of competition between iflavirus and SeMNPV? The authors should design experiments to study their interaction or competition at the level of virus infection, gene expression, or other mechanisms (such as the role of viral suppression of cell-mediated RNA silencing or host immune responses).
  3. Line 219: SeIV-1 treatment at concentrations of 107 - 109 genomes/μl resulted in a 100- to 10000-fold increase in viral titers. Since the unit is μl, these titers seem to be too high. The authors may have to recheck their results.

Author Response

Comments:

  1. Although these experiments are well-designed and the authors provide reasonable conclusions, the results do not significantly increase our knowledge of virus/host interactions or virus/virus competition in hosts.
  2. What are the competitive mechanisms between SeIV-1 and SeIV-2, and what is the mechanism of competition between iflavirus and SeMNPV? The authors should design experiments to study their interaction or competition at the level of virus infection, gene expression, or other mechanisms (such as the role of viral suppression of cell-mediated RNA silencing or host immune responses).

Response 1: The questions raised by the reviewer are of great interest and could be addressed in subsequent studies. The research mentioned by the reviewer is beyond the scope of the present study. We have mentioned this in lines 347-349. 

  1. Line 219: SeIV-1 treatment at concentrations of 107 - 109 genomes/μl resulted in a 100- to 10000-fold increase in viral titers. Since the unit is μl, these titers seem to be too high. The authors may have to recheck their results.

Response 2: Iflavirus treatments involving 107 - 109 genomes/μl were applied to egg masses, although the precise dose of particles consumed by neonate larvae during hatching is unknown. The viral titers mentioned in the manuscript are based on the quantification of samples comprising the whole insect (larvae and pupae) or the adult abdomen (lines 121-122). We confirm that the values mentioned in the manuscript have been checked and are correct.